# The Importance of Mitochondrial Pyruvate Carrier in Cancer Cell Metabolism and Tumorigenesis

**DOI:** 10.3390/cancers13071488

**Published:** 2021-03-24

**Authors:** Ainhoa Ruiz-Iglesias, Santos Mañes

**Affiliations:** Department of Immunology and Oncology, Centro Nacional de Biotecnología (CNB/CSIC), 28049 Madrid, Spain; aruiz@cnb.csic.es

**Keywords:** MPC, SLC, mitochondrial matrix, glycolysis, Warburg effect, oxidative phosphorylation, lactate

## Abstract

**Simple Summary:**

The characteristic metabolic hallmark of cancer cells is the massive catabolism of glucose by glycolysis, even under aerobic conditions—the so-called Warburg effect. Although energetically unfavorable, glycolysis provides “building blocks” to sustain the unlimited growth of malignant cells. Aberrant glycolysis is also responsible for lactate accumulation and acidosis in the tumor milieu, which fosters hypoxia and immunosuppression. One of the mechanisms used by cancer cells to increase glycolytic flow is the negative regulation of the proteins that conform the mitochondrial pyruvate carrier (MPC) complex, which transports pyruvate into the mitochondrial matrix to be metabolized in the tricarboxylic acid (TCA) cycle. Evidence suggests that MPC downregulation in tumor cells impacts many aspects of tumorigenesis, including cancer cell-intrinsic (proliferation, invasiveness, stemness, resistance to therapy) and -extrinsic (angiogenesis, anti-tumor immune activity) properties. In many cancers, but not in all, MPC downregulation is associated with poor survival. MPC regulation is therefore central to tackling glycolysis in tumors.

**Abstract:**

Pyruvate is a key molecule in the metabolic fate of mammalian cells; it is the crossroads from where metabolism proceeds either oxidatively or ends with the production of lactic acid. Pyruvate metabolism is regulated by many enzymes that together control carbon flux. Mitochondrial pyruvate carrier (MPC) is responsible for importing pyruvate from the cytosol to the mitochondrial matrix, where it is oxidatively phosphorylated to produce adenosine triphosphate (ATP) and to generate intermediates used in multiple biosynthetic pathways. MPC activity has an important role in glucose homeostasis, and its alteration is associated with diabetes, heart failure, and neurodegeneration. In cancer, however, controversy surrounds MPC function. In some cancers, MPC upregulation appears to be associated with a poor prognosis. However, most transformed cells undergo a switch from oxidative to glycolytic metabolism, the so-called Warburg effect, which, amongst other possibilities, is induced by MPC malfunction or downregulation. Consequently, impaired MPC function might induce tumors with strong proliferative, migratory, and invasive capabilities. Moreover, glycolytic cancer cells secrete lactate, acidifying the microenvironment, which in turn induces angiogenesis, immunosuppression, and the expansion of stromal cell populations supporting tumor growth. This review examines the latest findings regarding the tumorigenic processes affected by MPC.

## 1. Introduction

Cells monitor the availability of nutrients and oxygen in their microenvironment, and make metabolic adjustments to help them better meet their energetic needs. This ability to adapt to environmental factors is known as metabolic flexibility. Energetic plasticity is necessary for the self-renewal of stem cells, as well as for their entering the quiescent state, and for the differentiation of specific lineages. Stem cell differentiation is characterized by dynamic changes in carbohydrate metabolism, with a shift from glycolysis to mitochondrial-driven oxidative phosphorylation (OXPHOS). The high energy demand of rigorous physical activity also requires metabolic flexibility. In skeletal muscle, very intense exercise lasting less than one minute promotes glycolysis, with phosphocreatine and glycogen used as substrates to produce lactate and rapidly generated ATP—although this induces metabolic acidosis [1,2,3]. Exercise lasting longer than one minute enables OXPHOS as the major ATP-generating pathway [4]. Neoplastic cells, however, show metabolic inflexibility [5]. Indeed, an altered energy metabolism is a hallmark of different cancers [6].

The reprogramming of glucose metabolism from OXPHOS to aerobic glycolysis during oncogenesis—so-called Warburg effect [7,8]—is well known. This is counterintuitive from a bioenergetic point of view since glycolysis is far less effective than OXPHOS at generating ATP. However, glycolysis provides biosynthetic intermediates that are required by cancer cells for growth. In some cancers, these metabolic adaptations are irreversible due to somatic mutations, deletions, duplications, etc. of the genes coding for metabolic enzymes or their regulators. Many of these irreversible alterations directly affect the tricarboxylic acid (TCA) cycle [9], leading to increased glycolytic flow. An example is seen in the mutation of the isocitrate dehydrogenase (IDH1 and IDH2) genes in glioblastomas [10]. Theoretically, the majority of metabolic changes in cancer cells should be reversible, which in turn should cause a loss of their oncogenic properties. Metabolism-directed cancer therapies could target these reversible, and perhaps also irreversible changes, killing cancer cells (which are metabolically inflexible) while allowing non-transformed (metabolically flexible) cells to escape their effects.

Pyruvate is a critical compound in the above oncogenic switch since it is the crossroads between OXPHOS and lactic acid fermentation; it therefore determines the metabolic fate of glucose (Figure 1), and whether healthy or cancer-type metabolism is pursued. Many of the proteins that participate in pyruvate metabolism are differentially regulated in cancer and normal cells. To limit pyruvate oxidation, many tumors downregulate the mitochondrial pyruvate carrier (MPC) complex which transports pyruvate from the cytosol into the mitochondrial matrix. Consequently, the alteration of the MPC subunits, namely MPC1 and MPC2, determines the proportion of pyruvate used in lactic acid production (which occurs in the cytosol) or in OXPHOS. The downregulation of MPC1 and MPC2 has been associated with a pro-tumorigenic phenotype and in many cancers a poor clinical outcome [11]. In other tumors, however, it is thought that MPC upregulation might contribute to oncogenic progression [12,13,14,15]. This review examines what is known about MPC, its physiological function, and the consequences of positive and negative MPC deregulation in the different stages of oncogenesis.

## 2. Regulation of Pyruvate Levels in the Cytosol

Pyruvate is a key branch point for cellular metabolism since it bridges glycolysis and mitochondrial OXPHOS. Analyses of the relationships between respiration and glycolysis in neoplastic cells under aerobiosis revealed metabolic dysregulation that impairs pyruvate oxidation in the mitochondria. One mechanism behind this metabolic scenario is the negative regulation of mitochondrial pyruvate transporters, but the enzymes influencing pyruvate availability in the cytosol may also have a direct impact on the mitochondrial transport of this metabolite.

Pyruvate can be formed by the fermentation of glucose via glycolysis. In particular, it appears via irreversible transphosphorylation between phosphoenolpyruvate (PEP) and adenosine diphosphate, a reaction catalyzed by pyruvate kinase (PK) [16]. In mammals there are four different isoforms of PK: liver-, red blood cell-, and muscle-type, which exists as two isozymes (PKM1 and PKM2). The PKM1 isoform is expressed in terminally differentiated tissues that require a large supply of ATP, which is consistent with its ability to efficiently convert PEP to pyruvate. Unlike PKM1, which exists only as a tetramer, PKM2 forms tetramers (high affinity, low Michaelis constant for PEP) and dimers (low affinity, high Michaelis constant for PEP) [17]. In tissues and cells with high anabolic profiles, such as cancer cells, PKM2 is mainly found in the less active dimeric form. This causes the accumulation of glycolytic intermediates and their diversion to other anabolic pathways, such as the pentose phosphate pathway, which produces nucleotides involved in DNA replication, and the biosynthesis of serine, an allosteric activator of PKM2 (reviewed in [18,19]). Dimeric PKM2 has other activities beyond its canonical enzymatic function, such as the regulation of gene expression and protein kinase activity. Indeed, PKM2 phosphorylation or acetylation in the presence of serine triggers its entry into the nucleus, inducing—both directly and indirectly—the transactivation of hypoxia inducible factor-1α (HIF-1α) [20,21]. This in turn is a trans-activator of the PKM2 promoter [22]. Even the tetrameric PKM2 induces HIF-1α upregulation upon interaction with the dioxygenase/demethylase Jumonji domain containing 5 protein [23]. Finally, nuclear PKM2 induces Thr11 phosphorylation of histone H3, which enables the acetylation, inactivation, and removal of histone deacetylase 3 from the cyclin D1 and c-myc promoters, thus inducing their expression [24]. It is notable that both HIF-1α and c-myc induce the expression of a large number of glycolytic genes upon binding to highly conserved carbohydrate response elements (reviewed in [19]), boosting a feed-forward cycle that reinforces the glycolytic program in cancer cells.

Cytosolic pyruvate levels are also regulated by lactate dehydrogenase (LDH), which catalyzes the reversible reduction of pyruvate into L-lactate with the concomitant oxidation of NADH to NAD+. This NAD+ is needed for the continuous generation of ATP in cells that rely on glycolysis, allowing their survival even under anaerobic conditions. LDH is actually a family of at least six L-isomer-specific isoenzymes (LDH1-5 and LDH6/LDHX), coded for by three different genes: *LDHA* (muscle, M), *LDHB* (heart, H), and *LDHC* (testis; T). A fourth gene, *LDHD*, codes for a D-isomer-specific enzyme (reviewed in [25]). The products of *LDHA* and *LDHB* can combine in five different homo- or heterotetrameric forms: LDH-1 (4H), LDH-2 (3H1M), LDH-3 (2H2M), LDH-4 (1H3M), and LDH-5 (4M). Despite their strong structural similarity, LDH isoenzymes show significant differences in the charged residues surrounding the active site [26]. This difference determines the enzyme-substrate (pyruvate or L-lactate) binding affinity, and consequently the reaction catalyzed (LDH-A isoenzymes convert pyruvate into L-lactate whereas LDH-B catalyzes the reverse reaction). Thus, the profile of the LDH isoforms influences pyruvate degradation or synthesis.

In many aggressive cancers, the LDH-5 isoform is upregulated, probably as a consequence of the transactivation of the *LDHA* promoter by transcription factors responsible for the metabolic rewiring of their cells, such as c-myc [27] and HIF-1α [28]. This leads to the rapid transformation of pyruvate into L-lactate, reducing the pool of cytosolic pyruvate entering the mitochondria. Unlike that seen for *LDHA*, metastatic cancers usually show reduced *LDHB* expression due to promoter hypermethylation or altered glycolytic signaling; low LDHB levels have been associated with poor prognosis in different cancers (reviewed in [25]). Congruently, *LDHB* expression is a marker of response to neoadjuvant chemotherapy in breast cancer [29]. The anti-tumor activity of *LDHB* expression is, however, not universal. For example, strong *LDHB* upregulation is a predictor of poor survival in KRAS lung tumors and triple negative breast cancers [30,31]. The role of *LDHB* in oncogenesis seems therefore to be context-dependent.

## 3. Metabolism of Pyruvate in the Mitochondria

Under aerobic conditions, pyruvate is mostly transported into the mitochondrial matrix, where it is metabolized by the enzymes of the TCA cycle, with the ensuing production of ATP via the electron transport chain. In low oxygen environments, pyruvate stays in the cytosol and (i) is converted to lactate, which is then exported from the cell (the most efficient way to ensure NAD+ recycling), or (ii) is used in other biosynthetic reactions.

A limiting step of pyruvate metabolism is its transfer from the cytosol to the mitochondrion. Mitochondria have functionally different outer (OM) and inner (IM) membranes that encapsulate the intermembrane space and the matrix, respectively. These compartments are involved in different processes related to oxidative metabolism, biosynthetic pathways, and signaling [32]. The exchange of metabolites across the above membranes therefore needs to be regulated. The OM is permeable to low-molecular weight molecules via voltage-dependent anion channels (VDAC) (or porins).

VDACs are 3 nm-diameter channels in the OM that allow the passage of molecules up to 5 kDa in size [33] and reviewed in [34], depending on their charge. They are not “open all hours” gates but are regulated by a voltage sensor. This voltage gating depends on many factors including the availability of small molecules such as NADH, colloidal osmotic pressure, the phosphorylation of the porins and their interactions with other proteins, etc. Thus, the permeability of these channels depends on the competition or synergy between multiple factors, broadly discussed in [35,36]. Their open or closed conformation has an impact on mitochondrial metabolism and cell energetics. In the closed state, small ions—but not most anionic metabolites, including ATP/ADP+Pi and pyruvate—can cross the OM through them. When they are open, pyruvate travels from the cytosol to the intermembrane space through these channels, a finding confirmed in a study of a VDAC-deficient patient who presented with poor pyruvate oxidation and ATP production rates [37].

In eukaryotic cells, VDACs may exist in three isoforms. VDAC1 and VDAC2 are the major ones found in mammalian cells, including cancer cells (90% of all). VDAC3 is abundant only in normal testis tissues, but is also found in about 10% of cancer cells [38]. Experimental evidence gathered using different combinations of VDAC isoform knockdowns showed that VDACs regulate the maintenance of mitochondrial metabolism and the intracellular flow of energy [39]. In HepG2 cancer cells, VDAC1/2/3 knockdown reduces the potential of the mitochondrial membrane. This happens particularly with the knockdown of VDAC3, and when the NAD(P)H/NAD(P)^+^ ratio, ATP and ADP levels, and total adenine nucleotides, are reduced [40]. The latter authors also demonstrated that free tubulin closes the VDACs, impairing conductance and reducing the activity of the adenine nucleotide translocator (ANT), thus contributing to the suppression of mitochondrial metabolism and low cytosolic ATP/ADP ratios in cancer cells [41].

VDAC was also shown to be associated with the OM enzyme hexokinase II [42,43], which is overexpressed in tumor cells and required for tumor initiation and maintenance in murine models of cancer. It binds to VDAC1, inducing its closure and blocking the opening of the mitochondrial permeability transition pore (MPTP), which are pores that releases the pro-apoptotic protein cytochrome C [44]. The defective opening of MPTP prevents key events in mitochondria-mediated apoptosis. Hexokinase binding to VDAC also favors glycolysis [42,43].

Other regulatory mechanisms also exist, such as post-translational modifications of VDAC by protein kinase A (PKA) and glycogen synthase 3β (GSK3β) that reduce and increase conductance, respectively [45,46].

The IM, in contrast, is an impermeable barrier that allows only the flow of certain metabolites via specific transporters or mitochondrial carriers (MCs) [47]. The majority of MCs belong to the canonical mitochondrial carrier family (solute carrier family 25, SLC25) [48]. The transport of pyruvate across the impermeable IM, however, is undertaken by the non-canonical MPC [49,50].

Once inside the mitochondrial matrix, pyruvate has several potential fates. It can be used in the citric acid cycle to support ATP generation by OXPHOS, or be converted into glycerol, fatty acids, or amino acids. Along with the availability of pyruvate transporters, its fate is determined by the inhibition of the pyruvate dehydrogenase complex (PDC) by pyruvate dehydrogenase kinases (PDK) or by pyruvate carboxylase (PC), the activity of which correlates with gluconeogenesis [51]. PDC is a multi-enzyme complex located in the mitochondrial matrix that catalyzes the NAD^+^- and CoA-dependent decarboxylation of pyruvate to acetyl-CoA [52]. The resulting acetyl-CoA can continue being oxidized in the TCA cycle for further ATP production, or be used for fatty acid and cholesterol synthesis. Each complex is formed by multiple copies of three enzymes, E1, E2, and E3 and the e3 binding protein (e3BP) in octahedral or icosahedral symmetry, with E1 the rate-limiting enzyme. PDCs are characterized by a mobile swinging domain that provides for high substrate specificity. It also enhances reaction rates via the integration of the active sites of all three enzymes (reviewed in [53]).

The inhibition of PDC influences pyruvate availability for NADP by recycling LDH, by replenishing the TCA cycle with intermediates via the action of pyruvate carboxylase (PC), and via the transamination of pyruvate by alanine aminotransferase. This invests PDC regulators with a critical role in pyruvate metabolism, and thus cellular energy production and anabolic metabolism, as collected in [54]. PDC activity is controlled at different levels. At high concentrations, for example, small molecules such as ATP, NADH, or acetyl-CoA are inhibitory [55]. It is also controlled at the transcriptional level. For example, in the fasted state, transcripts for PDC enzymes are much fewer in number, while in the well-fed state there are many more [56]. Finally, rapid regulation of the complex is achieved thanks to kinases and phosphatases, which in turn are under allosteric and transcriptional regulation. PDK isozymes phosphorylate specific serine residues in the E1 alpha subunit of PDCs, inactivating them [57], allowing three carbon molecules to be used for the production of glucose. Conversely, pyruvate dehydrogenase phosphatase (PDP) reactivates PDC by dephosphorylation. Another post-translational modification of PDC involves the acetylation of the E1 alpha subunit by acetyl-CoA acetyltransferase 1 (ACAT1) (which can be reverted by SiRT3) [58,59]. Acetylation results in PDK recruitment and the inhibition of PDC.

PDKs are upregulated in metabolic diseases such as obesity, diabetes, heart failure and cancer [60,61,62]. Since PDC inhibition conserves substrates for cellular growth, the idea of using specific PDK inhibitors has been studied for treating patients with these problems. In breast cancer cells, the tumor suppressor p53 represses PDK2 transcription, removing its inhibitory effect on PDC [63]. Hypoxia is also a major inducer of PDK1 via an HIF-1α-dependent mechanism [64,65], which reinforces the glycolytic program induced by this transcription factor. Dichloroacetate, which inhibits this PDK isoform, shifts cancer cell metabolism towards OXPHOS; cancer cells thus-affected may enter ROS-dependent apoptosis by p53 activation and HIF-1α inhibition, reducing tumor growth [66,67].

The mitochondrial matrix enzyme PC also acts on pyruvate, transforming it into oxalacetate in an ATP-dependent manner—an anaplerotic reaction since the product is a recycling intermediate of the TCA cycle. This replenishment is critical for the complete oxidation of acetyl-CoA, as well as pathways that begin with intermediates of the cycle. Active PC is a tetramer formed by two dimers, which in turn have three functional domains: a biotin carboxylase (BC), a carboxyltransferase (CT), and a biotin carboxyl carrier protein (BCCP) domain. Acetyl-CoA and ATP are positive allosteric effecters of the activity of the enzyme after binding to the BC and CT domains of the protein. Glutamate, in contrast, is a negative allosteric regulator [68,69]. PC is also subjected to transcriptional control at the two promoters of the *PC* gene [70,71]. In glioblastoma cells, Cheng et al. showed that PC is induced by interruptions in glutamine metabolism, establishing PC as sufficient for glutamine-mediated anaplerosis in this tumor [61]. Other tumor types, such as lung, breast, and liver tumors, show constitutive PC expression [62,72,73].

## 4. Structure of MPCs

Evidence suggesting the existence of specific pyruvate transporters in the IM dates back to the 1970s [74,75]. In 1971, Papa et al. showed pyruvate transport to be associated with proton or hydroxyl ion exchange [74]. Pyruvate can cross membranes passively if its protonation state is favorable [76,77]. Under physiological conditions, the transport of pyruvate requires a pH gradient from the cytosol to the mitochondria [74]. This renders it very sensitive to changes in mitochondrial matrix pH. Using an inhibitor-stop technique, Halestrap studied the kinetic variables that determine the K_M_ and V_MAX_ of MPC, and the activation energy it requires [78]. It was later discovered that pyruvate transport occurs primarily with the symport of a proton, but not with the exchange of a hydroxyl ion [79]. However, MPC was not identified as an IM pyruvate transporter until 2012. Yeast and *Drosophila* mutants lacking the Mpc1 gene were found to have defects in pyruvate mitochondrial uptake, leading to reduced concentrations of acetyl-CoA and TCA [49], and *Mpc1* silencing in mammalian cells was seen to impair pyruvate oxidation. Independently, it was reported that MPC mutant yeasts cultured in valine- and leucine–free media showed reduced growth, reflecting a malfunction in the synthesis of lipoic acid, a derivative of mitochondrial pyruvate [50].

MPC belongs to the SLC54 family of mitochondrial transporters, which are highly conserved from yeasts to humans [80]. Initial blue native gel electrophoresis experiments indicated that MPC complexes had a MW of 150 kDa or higher [15], suggesting that they are multimers formed by various subunits. Recently, it has been possible to purify and reconstitute functional MPC heterocomplexes from *Saccharomyces cerevisiae*, which showed that the functional unit of the MPC complex is a heterodimer formed by different MPC protomers in a 1:1 ratio [22]. This study also confirmed the pH-dependency of pyruvate transport. In yeast, there are three MPC proteins, Mpc1, Mpc2, and Mpc3, which can form the heterodimers MPC1/2 and MPC1/3 [81]. MPC1/2 complexes are assembled under fermentative conditions, as seen in rapidly proliferating cells, and MPC1/3 forms under respiratory conditions [82,83]. It is notable, that MPC1/3 complexes transport pyruvate more efficiently than do MPC1/2 heterodimers, which depend on the C-terminal region of MPC3 for their activity [82]. These results strongly support the idea that the regulation of pyruvate import into the mitochondrion is a major factor influencing the metabolic switch associated with specific cell fates.

Unlike in yeasts, in mammalian cells, MPC complex activity seems not to be regulated by changes in subunit composition. Human MPC is formed by heterodimers of two proteins, MPC1 (SLC54A1, 12.3 kDa) and MPC2 (SLC54A2, 14.3 kDa), although in placental mammals a paralog of MPC1—MPC1L (SLC54A3, 15.1 kDa)—also exists [84]. The deletion of one MPC isoform results in the degradation of the other, leading to the complete failure of MPC to transport pyruvate into the mitochondria [85]. It is worth noting, however, that Nahgampalli et al. reported the functionality of human MPC2 oligomers on their own [86]. The presence of high order oligomers of MPC2 was observed during cryoelectron microscope examinations of MPC2-EGFP chimeric protein reconstituted in styrene maleic acid lipid particles, although the dominant size of the EGFP tag precluded any conclusion being drawn on the structural organization of the MPC2 oligomer [87]. Using a yeast homologous expression system, however, Tavoulari et al. concluded that MPC homodimers, although they can form, are non-functional [81]. More recently, Lee et al. [85], using a baculovirus expression system, confirmed the presence of homo- and heterotypic interactions between human MPC protomers, but concluded that heterodimers are the more stable and efficient at transporting pyruvate than monomers.

Based on homology analyses with other transmembrane proteins, it was suggested that the MPC1 and MPC2 protomers have different topological features. MPC2 is predicted to have three α-helical transmembrane domains (TM1-3), a short helix in the loop between TM1 and TM2, an N-terminal amphipathic α-helix facing the mitochondrial matrix, and a C-terminus orientated towards the intermembrane space [88]. In contrast, MPC1 has only two TM domains, and the N- and C-terminus domains both face towards the matrix [82]. Functional MPC1:MPC2 heterodimers thus consist of only five TM regions, not the minimum of six TM regions needed to form a pore in other mitochondrial transporters. MPCs are highly homologous to the bacterial SWEET (“sugars will eventually be exported transporters”) transporters [80], which show a 3 + 1 + 3 transmembrane architecture. The SWEET proteins can also be found as half-transporter homodimers (SemiSWEET) that contain the transmembrane helix repeat [89]; this SemiSWEET transporter dimerizes to form a complex with six TM domains [90]. Bioinformatic approaches for detecting pore-lining regions in transmembrane proteins suggest that only the TM3 region in human MPC2 is involved in pore formation. None of the TM helices of MPC1 is pore-facing, suggesting that this MPC protomer does not contribute towards pore formation. If so, functional MPC complexes must be oligomers rather than heterodimers, in which MPC1 would only regulate the stability of MPC2 oligomers. A recent homology analysis indicated, however, that MPC1 might have a topology similar to that of MPC2 and MPC3 [81], and consequently MPC1:MPC2 heterodimers would have all six TM regions required for pore formation.

It is noteworthy that all human diseases associated with pyruvate transport defects are linked to point mutations in MPC1, but not in MPC2. In the absence of MPC structural data, it is tempting to speculate that structural alterations affecting MPC1 transporter function caused by disease-inducing mutations might shed light on the functional link between pyruvate transport and MPC membrane organization [80].

The biogenesis of MPC proteins occurs in the cytosol, and consequently they must be transported to their destination in the inner mitochondrial membrane. This step is also far from being fully understood, although recent findings in yeasts suggest that MPC proteins are imported via the mitochondrial import pathway, which involves Tom70, small TIM chaperones, and the TIM22 complex [91].

## 5. Physiological Functions of MPCs

The transport of pyruvate into the mitochondrion is a critical event in cellular homeostasis: if it occurs, oxidative phosphorylation ensues, if not lactic fermentation occurs to regenerate NAD^+^ (which is required in the glycolytic pathway). The genes involved in pyruvate metabolism are therefore tightly controlled [51,92]. Although pyruvate can be produced in the cytosol by different routes, MPC is the only carrier that can transport it into the mitochondrial matrix. MPC deficiency may only have a limited impact on cell metabolism, however, since other metabolic pathways, e.g., glutaminolysis or the beta-oxidation of fatty acids, can compensate in substrate provision. Indeed, a recent study showed that the pharmacological inhibition of MPCs in brown adipocytes leads to an increase in energy production via fatty acid oxidation [93]. The metabolic flexibility of normal tissues thus renders the importance of MPC dependent on the physiological context.

Many MPC functions were discovered through the analysis of spontaneous mutations. In humans, mutations affecting conserved Mpc1 amino acids were reported in three families suffering from lactic acidosis, hyperpyruvatemia, and pyruvate oxidation defects [49]. A child with a presumptive MPC deficiency died prematurely with hypotonia, mild facial dysmorphia, periventricular cysts, marked metabolic acidosis, and hyperlacticemia [94]. Homozygous *Mpc1* and *Mpc2* knockout (KO) mice die during embryonic life, while heterozygous deletions are associated with no overt phenotype [95,96]. Homozygous mice with N-terminal-truncated MPC2 have a milder disease phenotype characterized by elevated blood lactate. *Mpc1/2* silencing does not affect the viability of cultured animal cells because they can compensate by using other substrates such as glutamine as a TCA substrate [97]. All these phenotypes reveal the importance of tightly regulating MPC activity.

Carbohydrate oxidation can differ between organs; MPC malfunction therefore affects glucose homeostasis in an organ-dependent manner. In pancreatic β-cells, MPC is required for glucose-stimulated insulin secretion (GSIS), which reduces blood glucose levels [95]. The loss of MPC in mice abrogates GSIS, resulting in hypoinsulinemia, impeding the reduction of blood glucose and leading to glucose intolerance [98]. In contrast, MPC inhibition in the liver is associated with glucose tolerance due to an insulin-sensitizing effect (as if under high blood glucose conditions), increasing glucose uptake and reducing gluconeogenesis [99]. MPC loss in muscle also increases glucose tolerance by diverting cytosolic pyruvate into lactate, which reduces the amount of glucose oxidation that occurs, but increases glucose uptake [100]. Many studies showed that high *Mpc1/2* expression induces gluconeogenesis, whereas low MPC expression leads to hypoglycemia [101]. This explains why some MPC targeted treatments ameliorate glucose intolerance and insulin resistance in patients with type II diabetes [102].

Pyruvate and MPCs are particularly important in the central nervous system (CNS) as metabolism here relies mainly on glucose. In neurons, pyruvate is generated through glycolysis and by the conversion of astrocyte-produced lactate through the so-called astrocyte-neuron lactate shuttle [103,104]. Given the key function of MPCs in glucose homeostasis and the co-morbidity between neurodegenerative and chronic metabolic diseases, such as type II diabetes, it is not surprising that alterations in MPC activity were implicated in neurodegeneration. Lactate and pyruvate accumulate in the cerebrospinal fluid of patients with Alzheimer’s disease (AD) [105], and the flux of pyruvate via PDC is reduced in AD brains, although PDH protein levels are not altered [106]. This suggests that MPC activity is diminished in AD, with less pyruvate entering the mitochondrial matrix to be transformed by PDH. Curiously, the administration of MPC inhibitors to non-diabetic subjects with mild/moderate cognitive decline increases glucose uptake in specific regions of the brain, suggesting a neuroprotective effect [107]. Lactate and pyruvate are also increased in the blood of patients with Parkinson’s disease (PD) [108], another neurodegenerative disease showing co-morbidity with type II diabetes. However, MPC inhibition provides neuroprotection in PD by targeting the mTOR pathway [109], which prevents neuroinflammation [110] and protects primary neurons from death caused by glutamate excitotoxicity [111]. MPC inhibition might indirectly affect Ca^2+^ entry into the mitochondria, helping to explain these apparently paradoxical results in neurodegenerative disease [112].

In the heart, 10–40% of all ATP generated comes from pyruvate oxidation, and ~65% of it is used in contraction [113]. Any breakdown in pyruvate oxidation could therefore cause contractile dysfunction. Mice with cardiac *Mpc2* deletion have enlarged hearts and show a loss of contractile function [114]. Under ischemic conditions, the reduction in oxygenation of the myocardium causes a switch towards anaerobic metabolism, increasing lactate levels [115]. Ischemia might therefore be ameliorated by enhancing MPC activity. Certainly, PDC activation [116] and pyruvate administration during reperfusion [117] were shown to improve cardiac function. During heart failure, pyruvate oxidation is impaired [118], with a switch occurring to the fetal glycolytic program, a switch that might be promoted by reduced MPC expression [119,120]. Further, MPC ablation is sufficient to induce cardiac hypertrophy and heart failure, whereas MPC overexpression in cardiomyocytes attenuates drug-induced hypertrophy [121]. Finally, the cardiomyopathy caused by doxorubicin treatment in patients with lymphoma is associated with MPC inhibition [122].

## 6. MPC Activity in Cancer Cells

Cancer cell metabolism is essentially glycolytic and thus characterized by a high glucose uptake and the production of lactate even in the presence of oxygen (aerobic glycolysis). Indeed, the uptake of ^18^fluorodeoxyglucose is used to detect tumors via positron emission tomography. The glycolytic switch associated with oncogenesis is a consequence of the biosynthetic requirements imposed by uncontrolled cell growth and proliferation [123]; simply put, glycolysis is the fastest way to transform nutrients into structural intermediates needed for the *de novo* synthesis of nucleotides, amino acids, lipids, and other biomolecules. Moreover, as well as the production of ATP, cancer cells require large amounts of reducing equivalents such as NADPH [123], which are obtained from NADP^+^ via the oxidation of carbon sources in pathways other than mitochondrial electron transport. In addition, glycolysis transforms glucose to lactate rapidly, producing ATP more quickly than via the more complete oxidation that occurs in mitochondria, providing cancer cells with a selective advantage when competing with stromal cells for limited resources [124].

Since it is common for cancer cells to switch from oxidative to glycolytic metabolism, it may be that the control of pyruvate metabolism is involved in promoting the transformed phenotype [125]. Cancer cells show many alterations in their expression of pyruvate metabolizing enzymes, such as an upregulation of lactate dehydrogenase (LDH) [126] (which transforms cytosolic pyruvate into lactate), in glucose transporters (explaining the increase in glucose uptake), and in other proteins involved in glycolysis [123]. Moreover, PDC, which performs the first step of mitochondrial pyruvate oxidation, is inactivated [64,65]. In addition, the dimerization of pyruvate kinase M2 leads to its inhibition, preventing pyruvate formation in tumor cells [127]. The impairment of pyruvate synthesis and oxidation is further reinforced by the loss of MPC activity, which dramatically boosts the Warburg effect. However, MPC re-expression or overexpression increases pyruvate oxidation and reduces glycolysis, switching metabolism towards OXPHOS [12,128,129,130,131]. Many oncogenic processes related to tumor progression are shaped by MPC expression, which might explain the poor prognosis associated with MPC downregulation seen in many types of cancer.

### 6.1. Regulation of MPC Expression and Its Association with Tumor Progression

The loss of pyruvate entering the mitochondria in tumor cells has long been associated with malignancy [132,133], but ascribing this to alterations in the expression or function of specific proteins has not been easy. Different authors now indicate that the repression or deletion of *Mpc1* and *Mpc2* is common in cancer, explaining the correlation between low MPC expression and poor survival seen in some cancers [11]. In many cases, MPC downregulation occurs at the transcriptional level, either by the direct binding of transcriptional repressors, as in the chicken ovalbumin upstream promoter-transcription factor II (COUP-TFII)-induced repression of MPC1 in prostate cancer [130], or through epigenetic mechanisms such as those seen at work in pancreatic cancer involving histone lysine demethylase 5A [134]. In kidney cell carcinoma, MPC1 downregulation is due to the silencing of peroxisome proliferator-activated receptor-gamma co-activator (PGC)-1α, which induces MPC1 transcription via an estrogen-related receptor alpha (ERRα)-mediated mechanism [135]. PGC-1α also upregulates MPC1 in cholangiocarcinoma and breast cancer, but in these cases ERRα-mediated MPC1 transcription seems to be required for tumor progression [12,13,14]. The androgen receptor (AR) drives MPC2 transcription and increases pyruvate oxidation and lipogenesis, which seems to be important for the progression of castration-resistant AR^+^ prostate adenocarcinoma subtypes [15].

MPC activity can also be downregulated by post-transcriptional mechanisms. For example, the acetylation of lysine residues 45 and 46 in MPC1, or of lysines 19 and 26 in MPC2, was associated with reduced MPC activity in cancer and in the diabetic heart, respectively [136,137]. In hepatocellular carcinoma, the tumor suppressor p53 negatively regulates MPC function through the upregulation of PUMA which, upon phosphorylation by IκB kinase-β, disrupts MPC1/2 dimer formation [138].

#### 6.1.1. Tumorigenicity

The increased glycolysis in cells with impaired MPC activity correlates positively with tumorigenicity (Figure 2). Pancreatic and colorectal cancer cells showing the suppressed expression of MPC adopt a spindle shape and downregulate *CDH1,* while upregulating *FN1* [139] (both markers of epithelial to mesenchymal transition [EMT]), a process associated with the development of migratory and invasive properties. This suggests that the repression of MPC enhances EMT and the formation of metastases (Figure 2). In renal clear-cell carcinoma, high MPC1 levels impair invasion in vitro and tumor growth *in vivo*, and are associated with increased overall survival [140], while in prostate cancer, *Mpc1* KO cells show enhanced proliferative, migratory and invasive capacity [15,130,141]. Moreover, *Mpc1/2* expression is of prognostic value in this cancer type: in a study of 88 patients, it correlated negatively with UICC stage and lymph node metastases, and positively with overall survival [142]. In glioblastoma, data from The Cancer Genome Atlas and the Genotype Tissue Expression database revealed a negative correlation between *Mpc1* expression and overall survival and response to temozolomide [143]. This association was also observed in esophageal squamous carcinoma, cholangiocarcinoma, lung adenocarcinoma, and colorectal cancer [129,136,144,145].

#### 6.1.2. Cancer Cell Stemness

The contribution of MPC suppression to tumorigenicity might be also related to the gain of stemness capabilities (Figure 2). MPC loss induces the proliferation and expansion of the stem cell compartment in intestinal organoids; in contrast, its overexpression in *Drosophila* stops stem cell division [146]. In prostate and ovarian cancer, *Mpc1* KO cells show an increase in stemness markers [141]. Bensard et al. suggest that the impairment of pyruvate import into the mitochondria promotes stemness and proliferation in a manner similar to that elicited by the Wnt/β-catenin pathway, and that this scenario triggers the earliest steps of tumor initiation [131]. In their study, these authors used two mouse models of colon cancer: (i) tumor induction by azoxymethane and dextran sodium sulfate (AOM-DSS) in drinking water, and (ii) the heterozygous loss of *Apc* in intestinal stem cells (Apc^Lrig1 KO/+^). In the AOM-DSS model, the deletion of *Mpc1* in intestinal stem cells (Mpc1^Lrig1 KO^) increased the frequency of adenoma formation and the grade of tumor compared to that seen for *Mpc1* WT animals, linking MPC loss with a greater susceptibility to tumor initiation after oncogenic stimulation. In Apc^Lrig1 KO/+^ mice, *Mpc1* ablation had no significant effect on tumor size, grade, or proliferation, suggesting that *Apc* mutant tumors are already highly glycolytic and cannot be potentiated by *Mpc1* loss. Moreover, MPC overexpression completely blocks the oncogenic effects of *Drosophila Apc*-mutant clones, an indicator that metabolism is downstream of the oncogenic pathways induced by *Apc* loss in intestinal stem cells. The transcriptional analysis of colorectal tumors of Apc^Lrig1 KO/+^ and Apc^Lrig1 KO/+^ Mpc1^Lrig1 KO^ mice revealed no differences in their stemness profile, indicating that once *Apc* is lost, Mpc1 ablation does not affect the stemness gene expression program. Using the PANTHER and Database for Annotation, Visualization and Integrated Discovery (DAVID) tools, these authors identified an inverse correlation between *Mpc1* expression and the Wnt signaling pathway, which is in charge of maintaining stem cell identity [147]. This is important since *Apc* is a repressor of the Wnt/β-catenin pathway in human tumors [148].

#### 6.1.3. Resistance to Therapy

MPC expression has been associated with the efficacy of some therapies (Figure 2). Certainly, MPC-deficient cells are more resistant to radio- and chemotherapy in vitro [129,139,141], and it was reported that patients treated with temozolomide for glioblastoma showed poorer survival when their tumors expressed low levels of MPC1 compared to patients with MPC1-intact tumors [143]. Nevertheless, in certain circumstances, MPC inhibitors might be of clinical use. For instance, MPC1 inhibitors trigger local reoxygenation that sensitizes tumor xenografts to radiotherapy [128]. MPC inhibition activates glutamate dehydrogenase (GDH), redirecting glutamine to feed the TCA cycle (Figure 1), making cancer cells more dependent on glutamine metabolism. In this scenario the combination of MPC1 with GDH inhibitors form a lethal combination that was very effective in preclinical models of liver cancer [149]. MPC downregulation also inhibits the IFNγ antitumor response in colon cancer cells, while MPC overexpression promotes ROS production and increases IFNγ-induced apoptosis [150].

### 6.2. Lactate and Acidification of the Tumor Microenvironment. Indirect Effects of MPC Dysfunction

A consequence of the exacerbation of the Warburg effect in tumors is the massive generation of lactic acid that must be extruded from cells to avoid intracellular acidification and to maintain glycolytic flow. Tumor cells, moreover, need to maintain a relatively alkaline intracellular pH for optimal metabolic enzyme activity (reviewed in [151]). As a result, lactic acid concentration, which in normal tissues ranges from 1.5 to 3 mM, can be as high as 40 mM in the TME, whereas the pH of tumors can be as low as 5.6 compared to 7.4 for normal tissues [152]. The H^+^/Na^+^-exchanger NHE1 and the monocarboxylate transporters (MCT) are key molecules in these respects. NHE1 is a reversible antiporter that uses the Na^+^ gradient to extrude cytosolic H^+^ ions into the extracellular space. MCTs are proton-linked plasma membrane reversible symporters; thus lactate/pyruvate transport into or out of the cell is associated with the co-transport of protons (H^+^) [153]. The MCTs form a family of four members (MCT1-4) belonging to the SLC16 gene family, with isoforms showing different affinity for pyruvate (reviewed in [151]). In tumors, the dominant isoform is MCT-4, which is induced by HIF-1α and which shows a low Michaelis constant for pyruvate, ensuring the preferential transport of glycolytically-produced lactate [154]. The inhibition of MCT seems to induce the acidification of the cytosol, whereas the forced expression of MCT4 increases the intracellular pH and accelerates the glycolytic flux, supporting the idea that MCTs have a pivotal role in glycolytically active tissues. Nonetheless, since MCT-mediated lactate transport occurs simultaneously with H^+^, competition for free protons between the MCT transporters and NHE1 (and other systems upregulated in tumors in order to maintain an alkaline cytosol) might occur, which would compromise the extrusion of lactic acid from the cell. This competition could be bypassed through cooperative mechanisms between MCTs and other proton-producing enzymes operating in tumor cells. Candidates include the intracellular and extracellular carbonic anhydrases. These produce protons via the hydration of CO_2_, which enhances the transporter activity of MCTs in both transformed and non-transformed cells [155,156]. Curiously, this cooperativity occurs even with catalytically inactive carbonic anhydrase mutants [157]. Recent evidence indicates that carbonic anhydrase might use parts of its intramolecular proton pathway to function as an H^+^ antenna that gathers protons to the MCT transporters, thus facilitating lactate transport [158].

The following sections examine the effect of lactate accumulation and TME acidification on the function of endothelial, mesenchymal, and immune cells, and their implications in tumor malignancy (Figure 2).

#### 6.2.1. Induction of Aberrant Angiogenesis

Angiogenesis is a characteristic of tumor progression. It occurs because of the aberrant expression of pro-angiogenic factors, such as vascular-endothelial growth factor (VEGF). However, angiogenic tumors usually remain hypoxic since, despite the formation of new blood vessels, the vascular network is dysfunctional [159]. Low oxygen tension in the TME induces adaptive cellular responses driven by hypoxia-inducible transcription factors (HIF-1 and HIF-2). These transcription factors are heterodimers formed from the HIF-α and HIF-β subunits. Whereas HIF-β is constitutively expressed, the HIF-α isoforms are regulated by the HIF prolyl hydroxylase domain proteins (PHD1–3), which label HIF-α for proteasomal degradation under normoxia [160]. One of the genes upregulated by hypoxia, particularly by HIF-1α-containing heterodimers, is that which codes for VEGF. Thus, a vicious cycle forms in which hypoxia feed forwards abnormal angiogenesis [160]. Curiously, increasing the stability of the HIF-2α subunit leads to increased tumor perfusion and reduced hypoxia [161,162].

The overproduction of lactate in glycolytic tumors is also involved in the abnormalization of the tumor vasculature via the inhibition of PHD, which stabilizes HIF-1α and increases VEGF levels in tumor endothelial cells (TECs) and other cells of the TME [163]. Lactate also induces TEC migration and tube formation via the inactivation of IᴋBα in TECs through the NFᴋB/IL-8 pathway [164], and it leads to the recruitment of vascular progenitor cells and induces vascular morphogenesis in vivo [165]. Therefore, lactate is not only a driver of tumor angiogenesis, it induces vasculogenesis in the tumor.

#### 6.2.2. Lactate as Metabolic Fuel

The progression and clinical course of many cancer types rely on the interaction of cancer cells with cancer-associated fibroblasts (CAFs) [166]. Lactate is a major regulator of CAF activity, serving as a source of energy for the cells in the TME. It induces the expression of MCT-1 and LDH-B in CAFs, leading to lactate uptake and its conversion to pyruvate, covering their energetic demands [167]. CAFs are very abundant in the TME and are supportive of cancer cells by enhancing proliferation and extracellular matrix remodeling [168]. The lactate secreted by glycolytic tumor cells also serves as a substrate for other tumor cells. Indeed, there is a process of lactic symbiosis in which tumor cells under hypoxic conditions produce and secrete lactate, while other tumor cells under aerobic conditions take up this lactate for OXPHOS and ATP production. This phenomenon has been called the ‘reverse Warburg effect’. In three different tumor models, the impairment of this lactate flux by MCT1 inhibition provokes the death of the oxidative cancer cells through glucose starvation [169].

#### 6.2.3. Lactate and Immunosuppression

Lactic acidosis has an immunosuppressive effect on the TME, impairing the immunosurveillance of immunogenic tumors. It is well-established that the activation of effector T cells co-occurs with the metabolic switch from OXPHOS to glycolysis, in such a way that the activated T cells start to produce lactate [170]. In tumor-infiltrating lymphocytes (TILs), however, the large amounts of lactate in the TME hinder their secretion of lactate, disturbing these cells’ metabolism, proliferation, lytic granule exocytosis, and cytokine production [171]. In contrast to effector T cells, regulatory T cell (Tregs) metabolism relies largely on OXPHOS [170]. The inhibition of glycolysis linked to high lactate levels in the TME thus sustains the OXPHOS metabolic program of Treg cells, which is reinforced by increased nicotinamide adenine dinucleotide oxidation. High lactate levels thus allow Foxp3-mediated reprogramming to resist T cell proliferation in low glucose environments and inhibit their function [172]. This is an advantage for tumors since Tregs are positively correlated with tumorigenicity and poor prognosis by their maintaining peripheral immune tolerance [173].

Lactic acid and acidosis can have a negative impact on the activity of anti-tumor cells. It is well described that lactate concentrations over 20 mM cause apoptosis in T lymphocytes, NK, and NKT cells [152]. NK cell cytolytic activity is also negatively regulated by reducing their expression of activation receptor NKp46 and by inhibiting the production of perforin and granzyme B [174]. Low pH (6.5) suppresses T-cell effector function, including IL-2 and IFNγ production and T cell receptor activation. Interestingly, proton pump inhibitors (e.g., esomeprazole, a specific inhibitor of H^+^/K^+^ ATPase) delay tumor progression in mice in an immune-dependent manner by buffering the pH at the tumor site [175]. Finally, lactic acid activates the IL-23/IL-17 pathway inducing Th17 differentiation and local inflammation, which can promote tumor progression [176].

Tumor infiltration by TILs is a major factor in the immune-mediated control of cancer and response to therapy [162,177,178]. The blockade of LDH-A in a melanoma mouse model was reported to improve the efficacy of anti-PD-1-based therapies [179]. This was linked to high numbers of TILs and NK cells, suggesting that lactate is a negative regulator of immune cell infiltration. However, lactate also accumulates at sites of chronic inflammation, such as the synovial membranes in patients with rheumatoid arthritis. High lactate levels here inhibit CD4^+^ and CD8^+^ T motility via Slc5a12 and Slc16a1 transporters [180]. The latter authors also showed reduced T cell motility in vivo to be associated with the impaired activation of glycolysis downstream of CXCR3, the receptor of the IFNγ-induced chemokines CXCL9, CXCL10, and CXCL11, which play a major role in chemoattracting T cells to tumors [181]. Therefore, whereas lactate seems to impede T cell infiltration into tumors, it seems to retain pathological T cells in autoimmunity. Whether lactate works differentially in autoimmune and oncological disease has not been studied. However, it is possible that lactate does not interfere directly with T cell infiltration into tumors but that it causes a reduction in T cell numbers in the TME by inducing apoptosis. It is also possible that lactate interferes with T cell function by affecting the density of immune suppressive cells. Indeed, lactic acid signals skew macrophages towards the M2 phenotype by inducing the expression of *ARG1* [182], which supports tumor growth and inhibits antitumor T cell responses [183]. Lactate can also induce M2 macrophage polarization in an indirect manner by triggering ERK-STAT3 in breast cancer cells [184]. Lactate also blocks monocyte differentiation into dendritic cells (DCs), hindering their antigen-presenting ability and inhibiting their release of cytokines. It also promotes tolerogenic DCs that strongly express IL-10, but only weakly express IL-12 [185]. Its effect in B lymphocytes is yet be explored, but given the effect of acidification on antibody stability it might be expected to cause their degradation or aggregation [186].

It is worth mentioning that the accumulation of acid lactic and H^+^ in the TME is largely a consequence of the deficient perfusion of tumors [187]. It is remarkable that the normalization of the tumor vasculature is usually associated with enhanced anti-tumor responses and improved response to immunotherapy [188,189,190]. Whether this association between improved vascular function and anti-tumor immunity is linked to a reduction in lactic acidosis in the TME has not been explored.

## 7. Conclusions

Mitochondria are not only important for ATP generation, they also provide the cell with signals that allow it to respond to changes in the environment. Mitochondrial malfunction is therefore a problem for cell metabolism and homeostasis. The compartmentalization of reactions and molecules is crucial for mitochondrial function, separating the cytosol, the intermembrane space, and the matrix. The OM is very permeable, but transport through the IM is limited, requiring the use of specific carriers. MPC transports pyruvate from the cytosol to the mitochondrial matrix. This is a limiting step in metabolism since pyruvate is a crossroads where many biochemical pathways meet. The inhibition of pyruvate entry adjusts circulating glucose, glutamine and glutamate utilization, the pyruvate-alanine shuttle, and fatty acid oxidation. Aberrant pyruvate metabolism is linked to diabetes, cancer, and cardiovascular and neurodegenerative disease. Thus, the study of the pyruvate flux from the cytosol to the mitochondria may help us better understand and design treatments for these conditions. In many cancer types, MPC is transcriptionally and/or functionally downregulated, which contributes to the glycolytic switch in tumors. It is not surprising, therefore, that MPC expression is closely related to tumor onset and aggressiveness, and to resistance to therapy. MPCs should therefore be studied as a biomarker of prognosis and treatment efficacy, but also as a potential target for attacking tumors, either alone or in combination with other agents.

Although the function of MPC in the TME has not been directly studied, in cancer cells MPC downregulation or loss increases lactate production and secretion. The acidification of the TME in this manner helps tumor progression by inducing angiogenesis and fostering cell populations such as those of CAFs that support cancer cells. In addition, lactate causes local immunosuppression in the TME by skewing the metabolism and the function of anti-tumor effector cells, and by promoting the differentiation programs of suppressor cells. As mitochondrial pyruvate import alters lactate accumulation and secretion, it is reasonable to think that MPC fluctuations in cancer provoke similar changes. Further study is needed on MPC and its role in cancer, including its importance in tumor-stroma cell communication.

## Figures and Tables

**Figure 1 cancers-13-01488-f001:**
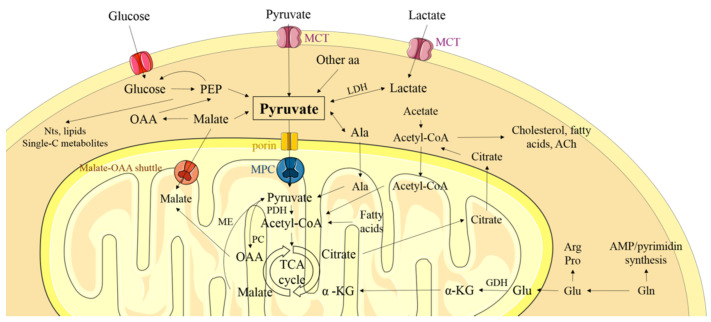
Pyruvate transport and metabolism in the cell. Pyruvate is derived from glucose, lactate, malate, and amino acids (aa). Under aerobic conditions, pyruvate enters the mitochondrion by crossing the outer membrane (OM) via porins, and the inner membrane (IM) via the mitochondrial pyruvate carrier (MPC) complex. Once inside, it enters the tricarboxylic acid (TCA) cycle to provide energy or intermediates for other biosynthetic reactions. Under hypoxic conditions, pyruvate is reduced to lactate in the cytosol, which is then secreted. PEP: phosphoenolpyruvate; LDH: lactate dehydrogenase; MCT: monocarboxylate transporters; α-KG: α-ketoglutarate; GDH: glutamate dehydrogenase.

**Figure 2 cancers-13-01488-f002:**
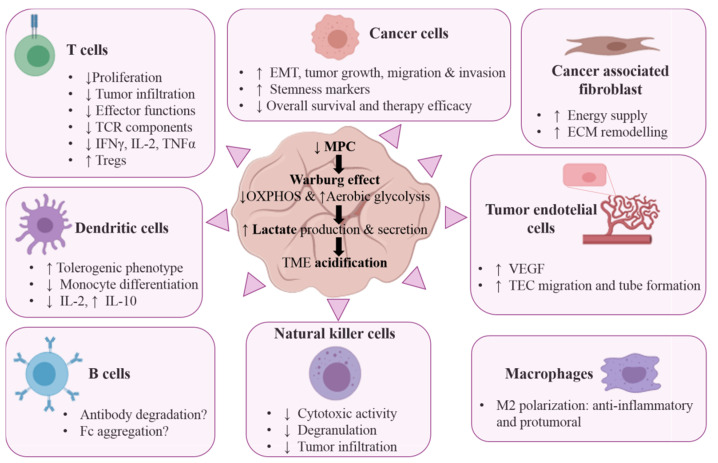
Overview of the impact of the Warburg effect caused by MPC loss, and the acidification of the tumor microenvironment (TME). Cancer cells undergo metabolic reprogramming from OXPHOS to aerobic glycolysis promoted by MPC loss/downregulation. This leads to the massive production of lactate, which is secreted into the TME, reducing the pH. These conditions promote the formation of new vasculature by vascular-endothelial growth factor (VEGF) signaling in tumor endothelial cells, favoring the growth of cancer-associated fibroblasts (CAFs) by providing lactate as a substrate for ATP production. They also affect different immune cell populations, causing immunosuppression.

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
