# Peer review of "The Importance of Mitochondrial Pyruvate Carrier in Cancer Cell Metabolism and Tumorigenesis"

_cancers, 2021, doi:10.3390/cancers13071488_

Round 1
Reviewer 1 Report
The review “ the importance of mitochondrial pyruvate carrier in cancer cell metabolism and tumorigenesis” is a very interesting review , well written and with an updated bibliography. Nevertheless, the authors missed a similar review published in Biomolecules on July 2020. The review entitled “the multifaced pyruvate metabolism: role of the mitochondrial pyruvate carrier” by Zamgari J., Petrelli F.,Maillot B. and Martinou JC. Biomolecules 2020 doi:10.3390/biom10071068 describes the role of the mitochondrial pyruvate carrier in cell metabolism and discuss the roles of dysfunctional MPC in several pathologies including cancer diseases. In this contest the novelity of the paper under revision is very poor; what I can suggest to the authors is to increase the difference between the Zamagni et al. review and their manuscript for example highlighing the role of VDAC in the pyruvate transfer from cytosol to mitochondria. For these reasons I think that this manuscript can be considered for publication only after major revision.
Author Response
Point-by-point rebuttal:
We appreciate the Reviewers’ insightful comments, which we believe have notably strengthened the review.
Reviewer 1:
We thank the reviewer for calling our attention on the interesting review by Zamgari et al (Biomolecules 2020), which certainly has some thematic overlap with ours. Nonetheless, whilst Zamgari’s review provided a more general view on MPC in various pathologies, we believe that the manuscript organization and the extension dedicated to cancer-specific MPC activities (particularly those related to the role of lactate and acidosis in the TME) make our review more focused and ‘attractive’ for oncologists, and hence suitable for Cancers.
In the revised version of the review, the referee will note that we have made substantial additions to the text, including an extensive discussion on the regulation of voltage-dependent anion channel-mediated transport of pyruvate, as suggested. We hope that with these new additions the referee finds our manuscript different from that published in Biomolecules.
Reviewer 2 Report
The MS of Ruiz-Iglesias and Manes is dealing with the role of mitochondrial pyruvate transporter (MPC) in pathological conditions with giving special emphasis to cancers.
The renewed interest in metabolism and the specific features of tumor metabolism make the current review very interesting and timely. The structure of the MS is logical, the review is well written, and references are adequate.
With a few comments and questions the referee thinks the paper can be even better.
Major points
1) Although the paper is about the role of MPC, this molecule is working in a metabolic environment, where its immediate neighbors can directly influence its function. Conclusions drawn from the study of one parameter could be misleading, especially if the parameter measured is very far from the phenomenon observed.
The cytosolic pyruvate and lactate level is greatly influenced by two enzymes, the pyruvate kinase (PK) and the lactate dehydrogenase (LDH). Both enzymes have several isoenzymes. Obviously the availability of pyruvate could be a major factor in the transport activity. Short discussion of the PK M2 would be worth, especially because slowing glycolysis at the 3 carbon atom stage at phosphoenol-pyruvate greatly contributes to the abundant formation of intermediates needed for the de novo synthesis of other indispensable molecules. Similarly, LDH isoenzymes exhibit important differences in their substrate affinity.
On the other hand on the mitochondrial matrix side two enzymes are playing a role in the utilization of entering pyruvate; the pyruvate dehydrogenase complex (PDHc) and the pyruvate carboxylase (PC). Both of them are regulated and their low activity could be rate limiting in the pyruvate flux. It would be nice to cite data about the vmax of MPC complex, PK, and PDHc and PC. (I think really more information would be required then that less than five sentence at the bottom of 5th page).
2) Stucture and function of MPCs
I also missed a more functional description of the MPC. Reading the text it is not clear whether the pyruvate is transported in the protonated or in the deprotonated form? Does the transport require the proton gradient? Could pyruvate be transported at low mitochondrial membrane potential?
3) Chapter 5.2.3. Lactate…
Considering that MPC can deeply influence the formation and release of lactate and vice versa it would be helpful to discuss the pH around the tumor cells, and its effect on the lactate transport.
4) Generally, I think it would be helpful to indicate in text whether a particular citation is a reference of a review or it is an original experimental study.
Minor points
1) 2nd page 1st paragraph “in skeletal muscle exercise promotes transition from OXPHOS to glycolysis.” This statement is not valid under many circumstances, e.g. physical exercise increases oxidative capacity stimulating PGC-1a transcription, therefore mitochondrial biogenesis is stimulated.
2) Question: which proton pump is inhibited by esomeprazole?
3) Chapter 6. Conclusion
The mTOR signaling pathway is mentioned in the conclusion, however in the MS it is mentioned only once in connection with Parkinson’s disease. Is it important to mention it in the conclusion?
Author Response
We appreciate the Reviewers’ insightful comments, which we believe have notably strengthened the review.
Reviewer 2:
Major points:
1) We concur with Reviewer’s arguments. We have added a new section 2 entitled “Regulation of pyruvate levels in the cytosol”, in which we discussed at a reasonable depth the main characteristics and regulation of PK and LDH isoenzymes, as key enzymes in the synthesis and catabolism of pyruvate. The PKM2 isoform and the functional differences between LDHA and LDHB isoforms are important aspects discussed in this section.
The old section 2 (Mitochondria and the metabolism of pyruvate) has been largely remodeled to accommodate the discussion on the regulation of pyruvate transport through the outer membrane by VDAC channels, as well as mechanisms that regulate pyruvate catabolism by PDC, PDK and PC isoforms, in a new section 3 entitled “Metabolism of pyruvate in the mitochondria”. When relevant, we also discuss kinetic differences between isoforms and transporters, and the relevance of these parameters in the metabolic flow. However we found certain dispersion in the Vmax and Km values provided in the literature; for this reason we do not include absolute values for these parameters.
2) We have tried to answer Reviewer’s concerns by rewriting some paragraphs and by adding new information on the structure-function studies performed on MPC. Since MPC structure has not been resolved yet, we have tried to highlight the controversies about the functionality of homo- and heterodimers, they oligomerization state, including new data obtained by cyo-microscopy. We also discussed briefly the importation of MPC to the inner mitochondrial membrane.
3) We thank the Reviewer for mention this important point. In the section 6.2, we have enhanced discussion on MCTs isoforms and their kinetics in cancer cells, examined the interrelationships between proton and lactate transport systems, as well as the cooperativity between MCTs and carbonic anhydrases.
4) We tried to indicate those references citing a review.
Minor points:
1) We apologize for the oversimplification of our statement, which have been now modified to better explain the metabolic rewiring in muscle during exercise.
2) The proton pump (H+/K+ATPase) is now indicated.
3) We concur with the reviewer and the mention to the mTOR signaling has been removed.
Round 2
Reviewer 1 Report
The revised version of the manuscript is improved and the added part contribute to differentiate this review from the Zamagni's one. in conclusion the manuscript can be accepted for the publication.